# Biocatalytic Method for Producing an Affinity Resin for the Isolation of Immunoglobulins

**DOI:** 10.3390/biom14070849

**Published:** 2024-07-14

**Authors:** Mikhail N. Tereshin, Tatiana D. Melikhova, Barbara Z. Eletskaya, Elena A. Ivanova, Lyudmila V. Onoprienko, Dmitry A. Makarov, Mikhail V. Razumikhin, Igor V. Myagkikh, Igor P. Fabrichniy, Vasiliy N. Stepanenko

**Affiliations:** 1Lomonosov Institute of Fine Chemical Technologies, MIREA—Russian Technological University, Vernadskogo Pr. 86, 119571 Moscow, Russia; misha060596@yandex.ru (M.N.T.); stepanenko_vasy@mail.ru (V.N.S.); 2Federal State Autonomous Educational Institution of Higher Education I.M. Sechenov First Moscow State Medical University of the Ministry of Health of the Russian Federation, 8-2 Trubetskaya Str., 119991 Moscow, Russia; youngchemist@mail.ru; 3Shemyakin-Ovchinnikov Institute of Bioorganic Chemistry, Russian Academy of Sciences, Miklukho-Maklaya St. 16/10, 117437 Moscow, Russia; tdm-63@yandex.ru (T.D.M.); onolv@mail.ru (L.V.O.); myagkikh@ibch.ru (I.V.M.); 4International Biotechnology Center “Generium” LLC, Vladimirskaya st. 14, 601125 Volginsky, Russia; lenchem98@ya.ru (E.A.I.); fabri@ibcgenerium.ru (I.P.F.); 5GreenVan LLC, Vodnikov Street, 16/2, of. 6, 125362 Moscow, Russia; razumikhin.m@gmail.com

**Keywords:** protein A chromatography, monoclonal antibodies, sortase A

## Abstract

Affinity chromatography is a widely used technique for antibody isolation. This article presents the successful synthesis of a novel affinity resin with a mutant form of protein A (BsrtA) immobilized on it as a ligand. The key aspect of the described process is the biocatalytic immobilization of the ligand onto the matrix using the sortase A enzyme. Moreover, we used a matrix with primary amino groups without modification, which greatly simplifies the synthesis process. The resulting resin shows a high dynamic binding capacity (up to 50 mg IgG per 1 mL of sorbent). It also demonstrates high tolerance to 0.1 M NaOH treatment and maintains its effectiveness even after 100 binding, elution, and sanitization cycles.

## 1. Introduction

Monoclonal antibodies (mAbs) have recently garnered considerable interest for therapeutic use. Approximately 100 mAbs have received marketing authorization from the USA Food and Drug Administration (FDA) until 2022, and the full list of antibody-based medications includes more than 500 cases [1,2,3]. The purity requirements for drug preparations containing antibodies have become more and more strict [4,5].

An effective technique for antibody capture is affinity chromatography on specific resins conjugated with bacterial proteins [6,7]. For example, the conservative antibody Fc region interacts with bacterial proteins A and G [8]; the Fab region of immunoglobulins (Ig) interacts with protein L, provided that the light chain belongs to the kappa isotype [9]. Protein A from *Staphylococcus aureus* exhibits a high affinity towards human IgG (IgG1, IgG2, IgG4 subclasses), which are mainly used as drugs for different types of therapy [5]. In this case, proteins A and G (or their fragments) are frequently employed as affinity ligands for resins in the process of mAb purification from different sources, including mammalian cell culture and transgenic plants [10,11,12].

The main disadvantages of affinity resins based on the natural protein A are the relatively high cost of synthesis, low binding capacity, limited number of life cycles, poor tolerance to alkaline conditions, ligand leakage, and requirement for low pH values during elution [12].

As previously shown, amino acid modifications in the domain B sequence of protein A can increase the pH values for IgG elution from 3.5 to 4.5 without specificity and efficiency loss [13]. Consequently, the B domain and its synthetic analogues are often used for the production of affinity resins for antibody isolation [14]. Additionally, several mutations resulted in increased stability of protein A-based resins under alkaline conditions [15,16,17]. The data presented by Scheffel et al. [18] show that the G28A replacement in domain B increases the tolerance of the polypeptide chain to hydroxylamine treatment.

Further modifications in the domain B structure resulted in the Zvar variant, containing the additional mutations Q9A, N11E, Q40V, A42K, N43A, and L44I. The affinity resin with the Zvar ligand was stable after six cycles of regeneration with 1 M NaOH solution, retaining 93% of its original capacity [16].

The crucial characteristic of the affinity resin is its dynamic binding capacity (DBC). Increasing the DBC value reduces the cost of mAb production. According to published data, the use of tandem repeats of protein A domains increases the dynamic capacity of resins [19]. In some cases, the introduction of an additional spacer between the matrix and the ligand allow for an enhancement of the DBC values of the resin by reducing steric hindrances during antibody binding [20].

Various approaches for peptide ligand conjugation on the solid matrices were described. Covalent conjugation through the functional groups of amino acids occurs nonspecifically, leading to a relatively low DBC [21,22]. Site-specific immobilization is more suitable for these purposes because it gives the possibility to control the spatial orientation of the protein and provides maximum ligand functionality. The appropriate orientation is achieved by the formation of a covalent bond between a certain ligand site (usually at the C- or N-terminus) and the functional group of the matrix [23,24,25,26].

A biocatalytic method for site-specific protein and peptide immobilization is an alternative to the approaches described above. It has been reported that the sortase A (SrtA) enzyme from *S. aureus* is able to attach proteins to various surfaces [27,28]. SrtA recognizes the LPXTG (X—any amino acid) sequence and cleaves it between threonine and glycine, resulting in thioacyl bond formation between LPXT and SrtA. Then, the N-terminal of an oligoglycine or primary amino group (attached to the matrix) nucleophilically attacks the thioacyl intermediate, leading to peptide bond formation between the substrate and the matrix (Figure 1) [27,28].

This research aimed to investigate an approach for the immobilization of the protein A analog, with the objective of developing affinity resins with improved capacity and stability and evaluating their application in isolating monoclonal antibodies (mAbs). To achieve this, an engineered mutant protein A with a sortase A recognition site and a C-terminal cysteine was designed and patented. Two methods for site-specific immobilization were then compared: chemical attachment via the C-terminal cysteine residue and enzymatic conjugation using sortase A. The obtained sorbents were tested using a purified antibody (bevacizumab) to determine the most effective approach to ligand immobilization. Then, the best sample was tested against two types of antibodies (bevacizumab and adalimumab) in comparison to the commercial resin MabSelect Sure (commercial protein A-based affinity resin). The properties of this resin were investigated under real conditions to purify the antibodies (types IgG1, IgG2, and IgG4) from the harvested cell culture fluid.

## 2. Materials and Methods

The following resins were used for modification: 6% cross-linked agarose beads Fast Flow (Sunresin, Xi’an, China); polymeric beads Seplite LX-1000EA (0.5 mmol amino groups/mL, Sunresin, Xi’an, China); 6% cross-linked Amino Sepharose 6 Fast Flow (17–22 mmol amino groups/mL, Cytiva, Uppsala, Sweden); and polymeric beads UniGel-65NH2 (65–70 mmol amino groups/mL, Suzhou NanoMicro Technology, Suzhou, China). The following IgGs were used for the testing of the affinity resins: HUMIRA^®^ (adalimumab from AbbVie, Ludwigshafen, Germany) and Avastin^®^ (bevacizumab from F Hoffmann-La Roche Ltd., Basel, Switzerland). All reagents from Panreac (Barcelona, Spain) were of analytical grade, unless otherwise indicated.

### 2.1. Recombinant BsrtA Protein and Sortase A Production

The production method for the recombinant BsrtA protein is described in patent RU No. 2789032 C1, published on 27 January 2023. The procedure involves synthesizing the B domain of the protein A analog, creating a recombinant plasmid for the expression of this protein, and utilizing an *Escherichia coli* strain capable of producing the corresponding protein [29,30]. A mutant Ca^2+^-independent sortase A with seven amino acid substitutions was obtained following a previously described procedure [31,32]. The purity of the isolated recombinant proteins was analyzed using electrophoresis and HPLC, and the results were >90% for both methods.

### 2.2. Epoxy Activation of Agarose Hydroxyl Groups with Diglycidyl Ether of 1,4-Butanediol (BDDE) or Epichlorohydrin (ECH)

Epoxy activation of sepharose was performed using epichlorohydrin and 1,4-butanediol (Figure 1) in accordance with the methods described in previously published papers [33,34].

The reaction was stopped by washing the epoxy agarose on a glass filter with a large amount of distilled water. The amount of introduced epoxy groups on the matrices was determined as follows. A total of 10 mL of 1.3 M Na_2_S_2_O_3_ solution and a couple of drops of an indicator (phenolphthalein) were added to epoxy agarose (1 g). The suspension was thermostated for 30 min at 30 °C, shaking occasionally. The sample was titrated with 0.01 M HCl solution until the indicator became discolored (pH = 7). The number of attached oxirane groups was determined by the amount of titrant solution used.

### 2.3. Preparation of Aminoethyl- and Aminobutyl-Sepharose

For the synthesis of the amino-agarose matrix (Figure 2), the method described in [20] was used.

Quantitative determination of amino groups was carried out by titration of Cl^−^ ions. Sample preparation was carried out as follows. A total of 3 mL of the matrix was placed on a Schott filter, successively washed with a solution of 1 M hydrochloric acid, and then washed with water until the pH of the eluate was neutral. The bound chloride ions were eluted from the carrier with 0.6 M sodium nitrate solution, and then the collected eluate was titrated with an aqueous solution of silver nitrate (0.1 M) in the presence of potassium chromate as an indicator. The equivalence point was determined when the yellow color of the solution changed to orange–red.

### 2.4. Preparation of Matrices with Polyglycine Ligands

A 10 g sample of any matrix with free amino groups was washed sequentially with water (5 × 20 mL), 20% ethanol solution in water (1 × 30 mL), 40% ethanol solution in water (1 × 30 mL), 60% ethanol solution in water (1 × 30 mL), and 100% ethanol (3 × 20 mL). Then, the washed matrix with amino groups (1 eq. amino groups, determined according to the titration method, see Section 2.3) was washed on a Schott filter with dimethylformamide (DMF). Polyglycine ligand solution 4 equiv. in relation to amino groups, HBTU (4 eq.), and ethyldiisopropylamine (8 eq.) in DMF was prepared and added to the sorbent. The reaction mixture was stirred for 2 h at room temperature (Figure 3). The preparation of polyglycine spacers is described in the Appendix A.

The course of the reaction was monitored using the Kaiser qualitative test [35] to determine the residual free amino groups. After the polyglycine ligand addition reaction was complete, the resulting matrix was washed with DMF using a Schott filter. To remove the Boc-protective group, the matrix with attached Boc-polyglycine was washed with 150 mL of distilled water, 15 mL of 95% trifluoroacetic acid (TFA) was added, and the reaction mixture was stirred for 2 h at room temperature. To remove TFA, the matrix was washed repeatedly on a Schott filter to neutralize the pH of the eluate. The effectiveness of removing the protective group was controlled using the Kaiser method.

### 2.5. Method for Determining the Amount of BsrtA Protein Ligated on a Matrix

The analysis was performed using the Pierce BCA Protein Assay Kit (Thermo Fisher, Rockford, IL, USA) method for a microplate. To obtain the calibration curve, 25 µL of a standard BsrtA protein solution in PBS was loaded into the microplate (from 1.5 to 0.2 mg/mL). The analyzed samples of sorbents with BsrtA ligated protein were washed with distilled water on a filter. Then, 400 µL of PBS was added to the suspension of 100 mg of the washed sorbent, the suspension was thoroughly mixed, and 25 µL was introduced into a microplate (the measurements were performed in 4 repeats). Then, 200 µL of a reagent mixture (BCA Protein Assay Reagent A + Reagent B 50:1) was added to the samples. The microplate was incubated at 37 °C for 30 min with shaking and incubated at room temperature (for 10 min). After sedimentation of the solid matrix particles in the reaction mixture, 125 μL of the supernatant was carefully removed from the microplate and transferred to another microplate. Measurements were performed on a Thermo Scientific Multiskan FC at 560 nm.

### 2.6. Chemical Immobilization of BsrtA through the Terminal Cys on Cross-Linked Agarose Beads 6%

Mercaptoethanol (10 μL) was added to 1 mL of protein solution (6.8 mg/mL). To change the buffer, the protein was loaded on Sepharose G25 in 100 mM NaHCO_3_ (pH 8.5) immediately before immobilization. Then, the protein was diluted with a buffer (100 mM NaHCO_3_, pH 8.5) to a concentration of 0.68 mg/mL and immediately added to the epoxy-activated cross-linked agarose beads synthesized according Section 2.2. The reaction was carried out for 4–8 h with stirring at room temperature. Protein immobilization was monitored spectrophotometrically in the solution (at 280 nm). To block the residual epoxy groups, 2 mL of 1 M ethanolamine solution in 100 mM NaHCO_3_ (pH 8.4) was added to the agarose 6% and incubated for 4–8 h at room temperature. At the end of the reaction, the agarose beads were washed sequentially with 100 mL of 100 mM NaHCO_3_ (pH 8.4) and 100 mL PBS (pH 7.4) solutions on a glass filter.

### 2.7. Enzymatic Immobilization of BsrtA Using Sortase A

The samples of matrices modified with different linkers were sequentially washed on a glass filter with the following buffer solutions (100 mL of buffer per 10 mL of a matrix): 100 mM NaHCO_3_ (pH 8.4); 50 mM Tris (pH 8.0); 100 mM sodium acetate (pH 4.5); and 50 mM Tris-HCl (pH 8.0).

A solution of BsrtA (9.5 mL of protein at a concentration of 21 mg/mL) in 50 mm Tris-HCl (pH 8.0) was added to 10 mL of a pre-prepared matrix. Then, 1.25 mL of 4 M NaCl and 15.3 mL of sortase A at 5.9 mg/mL concentration were added. The total volume of the reaction mixture was adjusted to 50 mL with 50 mM Tris (pH 8.0) buffer. The resulting molar ratio was the following: A total of 1 mL of the resin containing 65–70 µmol of amino groups was resuspended in 5 mL of a reaction mixture containing 1.28 µmol of BsrtA and 0.42 µmol of sortase A.

The protein concentration in the samples was evaluated using a UV/visible spectrophotometer Ultrospec-1000 (Amersham, Pharmacia Biotech, Cambridge, UK) spectrophotometrically, and the extinction coefficients were calculated with the ProtParam tool (https://web.expasy.org/protparam/, accessed on 1 September 2022) (ε^280^ = 4470 M^−1^cm^−1^ and 17,420 M^−1^cm^−1^ for B_srtA_ and sortase A, respectively). The reaction mixture was incubated with stirring at 37 °C.

After 3 h of the reaction, the sorbent was sequentially washed on a glass filter with the following solutions (100 mL of buffer solution per 10 mL of sorbent): PBS (pH 7.4); 100 mM sodium acetate (pH 4.5); PBS (pH 7.4); 100 mM NaOH; 50 mM Tris-HCl (pH 8.0); and an aqueous solution of 20% ethyl alcohol.

In parallel, a negative control experiment was performed without SrtA in the reaction mixture. The sample was incubated and then washed with various solutions (in accordance with the protocol) to eliminate any possible binding of BrstA due to non-specific interactions with the matrix. After washing, the sample was analyzed using the same method as the experimental sample, and the resulting DBC value was zero.

### 2.8. Dynamic Binding Capacity (DBC) Determination

The estimation of DBC_10_ values was carried out using a column C10/10 (Pharmacia, Bromma, Sweden, size 10 cm × 10 mm). The volume of the applied resin sample was 2 mL.

The linear flow velocity corresponding to a given residence time (1, 3, 6, or 10 min) was calculated using the following Formula (1):υ = (H/t) × 60,(1)
where t is the residence time of the IgG_1_ with the resin in min; H is the bed height of the resin in the column in cm; and υ is the linear flow velocity in cm/h.

The flow rate was calculated from the linear flow velocity using the following Formula (2):q = υ × S/60,(2)
where υ is the linear flow velocity in cm/h; S is the column cross-sectional area in cm^2^; and q is the flow rate in mL/min.

The flow rate during the washing and elution steps was 2 mL/min. To determine the DBC_10_ values, a working solution of immunoglobulin G_1_ 2 mg/mL in PBS (pH 7.4) was applied. The measurement of the maximum absorption of an IgG solution with a known concentration was carried out using a UV detector installed in the AKTA Pure instrument at 280 nm. After the resin was equilibrated with 10 mL of PBS (pH 7.4), the IgG solution was applied to the resin until an optical density equal to 10% of the original solution’s optical density was achieved. The dynamic capacity of the resin was calculated for the residence times of 1, 3, 6, or 10 min using the Dynamic Binding Capacity calculations in UNICORN Extension 7.3 SP1 software. Finally, the resin was washed with 10 mL of PBS (pH 7.4), and bound IgG was eluted with 100 mM sodium acetate (pH 3.5). After elution, the resin was washed with 10 mL of PBS.

### 2.9. CIP (Cleaning-in-Place) Procedure

The simulation of the sanitation of the resin was carried out by sequential washing with the following solutions: 10 mL PBS (pH 7.4); 10 mL 100 mM sodium acetate (pH 3.5); 10 mL PBS; 60 mL 100 mM NaOH; and 60 mL PBS. To determine resistance of the resin to CIP, DBC10 was measured after every 20 cycles at 3 min RT.

### 2.10. The Throughput of RUselect-P Resin at Various Flow Rates

The resin throughput was determined using a GE AKTA Pure chromatographic system with an XK16/40 column (GE Healthcare, Uppsala, Sweden) with a diameter of 1.6 cm. The column was packed with 30 mL of RUselect-P resin (the height of the column is approximately 15 cm).

### 2.11. Isolation of mAbs from the Harvested Cell Culture Fluid (HCCF)

HCCF samples containing IgG1, IgG2, and IgG4 antibodies at concentrations of 2–6 g/L were obtained by cultivation of CHO cell lines.

Chromatographic experiments were performed on an ÄKTA Avant system (GE Healthcare, Uppsala, Sweden) on prepacked HiTrap MabSelect SuRe columns with a volume of 1 or 5 mL (GE Healthcare, Uppsala, Sweden) and Tricorn 10/100 (GE Healthcare, Uppsala, Sweden) with RUselect-P resin (Affitrade, Moscow, Russia). The chromatographic process was monitored using a UV detector at 280 nm as well as pH and electrical conductivity sensors.

The protein concentration in the samples after purification was determined using a BioPhotometer plus spectrophotometer (Eppendorf, Hamburg, Germany) at 280 nm, based on the calculated absorption coefficients $A_{280}^{0.1\%}$ for the corresponding IgG. The obtained samples were analyzed by size-exclusion HPLC on the Alliance 2695/e2695 chromatographic system with UV/Visible 2487/2489 detector Waters, Milford, MA, USA) on a TSKgel G3000SWXL column (Tosoh, Tokyo, Japan) with a pre-column Security Guard Cartridge GFC 3000 (Phenomenex, CA, USA).

Quantitative determination of residual host cell proteins (HCP) in fractions was carried out by an indirect enzyme immunoassay using a test kit of our own production (Generium, Vladimir, Russia) based on goat antibodies to HCPs of CHO cells and their conjugate with horseradish peroxidase. BsrtA leakage was determined using the Protein A Mix-N-Go F610 kit (Cygnus Technologies, Southport, NC, USA). Measurements were performed on an XMark microplate spectrophotometer (BioRad, Hercules, CA, USA).

To determine the productivity of HCCFs, a 5 mL sample was loaded onto a MabSelect SuRe HiTrap column (5 mL CV) pre-equilibrated with PBS (pH 7.4); then, the column was washed with PBS (5 CV) and 50 mM sodium acetate (pH 5.5) (5 CV), followed by IgG elution with 50 mM sodium acetate (pH 3.5). The amount of a target IgG in the eluates was measured spectrophotometrically.

To evaluate the maximum capacity of the resins, such an amount of HCCF was loaded on the column so that the content of the target antibody in it (50–60 mg per 1 mL of sorbent) obviously exceeded the known DBC values for each resin (measured earlier for RUselect-P and declared by the manufacturer for MabSelect SuRe). The RT in each experiment was 2 min. Then, the column was washed with PBS (8 CV) and 50 mM sodium acetate (pH 5.5–6.0) (6 CV). Finally, an elution with 50 mM sodium acetate (pH 3.4–4.4) was performed.

The effect of RT on the maximum capacity was studied by loading HCCF samples on a column (60 mg of IgG per ml of the resin) at a flow rate corresponding to a given RT (2, 3, or 6 min). Washing and elution were performed at an optimal pH value for each resin and antibody.

To study the effect of washing with a high ionic strength solution on product purity, experiments were conducted with an additional 5 CV washing after loading the sample. PBS with 1 M NaCl (pH 7.3) or 50 mM sodium carbonate, 1 M NaCl (pH 10.0) was used as washing solutions.

## 3. Results

### 3.1. Design of the Recombinant BsrtA Ligand

In this study, we developed a new peptide sequence, designated BsrtA, derived from the B domain of protein A, as shown in Figure 2 [29,30]. The BsrtA design strategy involves a number of key features aimed at optimizing its functionality for various biotechnological applications. Notably, the sequence exhibits intentionally low affinity for Fab fragments, increasing its suitability for specific binding scenarios. Integration of the leader peptide improves the efficiency of peptide biosynthesis, while the inclusion of two fused B domains of protein A aims to increase the capacity of the resulting resin. Six introduced single mutations increase the tolerance of BsrtA to alkaline conditions during cleaning-in-place (CIP) processes. The inclusion of the LPETG sequence facilitates sortase A-mediated ligation to templates. In addition, the widely used His tag simplifies the isolation of the recombinant protein, and the inclusion of a cysteine residue at the C-terminus allows for chemical immobilization of the ligand on SH-functional matrices.

### 3.2. Influence of the Matrix Type and Conjugation Method on the DBC of the Resin

We studied several commercially available matrices based on agarose and polyacrylate/polymethacrylate (Table 1), with three of them (B, C, D) carrying amino groups and one (A) being amino-activated in our laboratory.

Covalent immobilization is a widely utilized technique for protein immobilization on solid matrices. Using the method of chemical immobilization of protein A described in the literature [19,31], we attached BsrtA to the agarose matrix type A via C-terminal cysteine (Sample 1 in Table 2). The effectiveness of the resulting resin was assessed based on a number of parameters, particularly the DBC10 (10% breakthrough dynamic binding capacity). The tests were performed using a solution of bevacizumab IgG (Avastin^®^, F Hoffmann-La Roche Ltd., Basel, Switzerland).

The maximum result achieved with chemical immobilization (DBC10 of approximately 25 mg of IgG1 per 1 mL of resin) was significantly lower than that of the commercial analogues, which typically have a DBC10 above 30 mg/mL at 3 min residence time (RT). The DBC10 value decreased by approximately 20% after 0.1 M NaOH treatment. Thus, we decided to explore an alternative biocatalytic method of immobilization. The sortase A heptamutant described in the literature was employed for this purpose [31].

According to the literature, the pentaglycine molecule is considered to be the optimal donor molecule for the transpeptidation reaction using sortase A [28]. Thus, the synthesized agarose matrix was modified with pentaglycines to evaluate the potential of enzymatic synthesis. BsrtA was immobilized on the matrix via sortase-mediated ligation, resulting in sample 2 (Table 2). The DBC_10_ value of sample 2 resin (enzymatic immobilization) was 30% higher compared to sample 1 (chemical immobilization). Moreover, sample 2 exhibited greater resistance to CIP.

To further optimize the enzymatic immobilization process, several samples were synthesized using various matrices and linkers. Taking into account the previous literature reports on the ability of sortase A to recognize not only oligoglycines [36,37] but also other compounds with a free amino group, immobilization was performed on commercially available matrices with primary amino groups (Table 3).

The resins synthesized on an agarose matrix coated with additional linkers (samples 2, 3, 4, and 5 in Table 3) exhibited higher DBC_10_ values compared to linker-free sample 6. We speculate that an additional linker between the ligand and the matrix may reduce steric hindrance. Samples 2 and 5 demonstrated the highest resistance to sanitation, indicating that the linker’s structure affects the tolerance of a resin towards alkaline conditions. In addition, two modifications of sample 2 were obtained using a tetraglycine or triglycine linker in place of pentaglycine. The resin with a tetraglycine linker showed a similar DBC_10_ value compared to sample 2. However, the use of a triglycine linker led to a 21% decrease in DBC_10_. Taking into account these results, we concluded that it is possible to use shorter oligoglycine linkers.

It is worth noting that immobilization was successfully performed using oligoglycine-free matrices (samples 6 and 9) coated by primary amino groups. Having measured DBC_10_ values, we found that the binding capacity of the resulting resins (ranging from 11.8 to 40.3 mg/mL) correlated with the number of free amino groups (ranging from 17 μmol to 65 μmol per 1 mL of the matrix for samples 6 and 9 micromoles, respectively). We attempted to improve these results by introducing an additional pentaglycine linker between a matrix and the BsrtA. The 5Gly-modified sample 5 exhibited a more than two times higher DBC_10_ value compared to the 5Gly-free sample 6 (27.1 and 11.8 mg/mL, respectively). The 5Gly-modified sample 8 and the 5Gly-free sample 9 showed similar DBC_10_ values (50.4 and 40.3 mg/mL, respectively). Considering that sample 6 was synthesized on the matrix type B with 17–22 μmol of amino groups per 1 mL of the matrix while sample 9 was synthesized on the matrix type D with 65 μmol of amino groups, we can conclude that high-yield biocatalytic immobilization of a protein ligand via sortase A could be carried out using a matrix with primary amino groups. However, a significant amount of amino groups seems to be crucial for achieving the best results.

Samples 2, 8, and 9 exhibited superior results in terms of capacity and stability after the CIP process. However, given the requirement for additional stages of the linker synthesis, our research focused on sample 9 (polymeric matrix with primary amino groups). By optimizing the conditions of enzymatic immobilization, we were able to increase the DBC_10_ value to 45.6 mg IgG1 per 1 mL of resin at 3 min RT. We named the sample 9 resin “RUselect-P”.

### 3.3. Characteristics of the Ruselect-P Resin

We evaluated the key characteristics of the Ruselect-P resin (Figure 3).

The DBC_10_ value for IgG_1_ bevacizumab (Avastin^®^, F Hoffmann-La Roche Ltd., Basel, Switzerland) exceeded 50 mg of IgG1 per ml of the resin at a RT of 10 min (Figure 3A). Commercial protein A resins demonstrate comparable DBC_10_ values [11,12]. The direct comparison of Ruselect-P with the commercial protein A-based resin MabSelect SuRe is presented below.

We studied the tolerance of Ruselect-P towards a 0.1 M NaOH solution and its stability after repeated use. The resin did not lose its capacity after 200 regeneration cycles (Figure 3B). This ability to maintain capacity during multiple regenerations is a significant advantage of affinity resins based on protein A. Intra-day and inter-day precisions for IgG1 bevacizumab were <1.7% and <2.0% RSD (relative standard deviation), respectively. The data were analyzed using Student’s *t*-test, and *p* < 0.05 was considered significant.

During the pilot-scale production of mAbs, affinity resins are often used at high flow rates, so we evaluated the column pressure at different velocities (Figure 3C). The operational properties of the resin make it possible to carry out the chromatography process at high flow rates: up to 500 cm/h (residence time of 1.5 min), the pressure did not exceed 0.45 Mpa under the experimental conditions.

DBC_10_ values for Ruselect-P and MabSelect SuRe (Cytiva, Uppsala, Sweden) were evaluated at different RTs (Figure 4). The experiment involved two commercial IgG_1_s: bevacizumab (Avastin^®^, F Hoffmann-La Roche Ltd., Basel, Switzerland) and adalimumab (Humira^®^, AbbVie, Ludwigshafen, Germany).

We found that the DBC_10_ values varied depending on the nature of the antibodies. The capacity of Ruselect-P at any RT is slightly higher when using bevacizumab, while MabSelect SuRe exhibited higher capacity when using adalimumab. Nevertheless, the Ruselect-P resin demonstrates DBC_10_ values comparable to those of MabSelect SuRe.

### 3.4. Testing Ruselect-P for IgG Isolation from Harvested Cell Culture Fluid (HCCF)

To determine whether the Ruselect-P resin could be used for the industrial production of antibodies, we carried out a series of experiments. Three antibodies belonging to the IgG_1_, IgG_2_, and IgG_4_ subclasses were purified directly from clarified HCCF using the Ruselect-P resin. Simultaneously, similar experiments were carried out with the widely used MabSelect SuRe resin. To determine the maximum resin capacity, the isolation process was initially performed under standard conditions for the MabSelect SuRe resin, involving a conditioning wash at pH 5.5 and elution at pH 3.2 and the 2 min RT. Under these experimental conditions, the amount of protein eluted from the Ruselect-P resin was found to be 15–35% lower than that of MabSelect SuRe, depending on the antibody subclass (Figure 5). This difference can be attributed to the fact that a larger amount of IgG was desorbed from Ruselect-P (up to 27% in the case of IgG_1_) than from MabSelect SuRe (up to 14% also for IgG1) during the conditioning wash with a pH 5.5 solution. This suggests that the affinity of antibodies to the Ruselect-P resin is slightly lower than that to MabSelect SuRe (and the difference is more pronounced for IgG1 and IgG2 mAbs than for IgG4), and a decrease in the pH value of the conditioning buffer to 5.5 leads to premature desorption of a significant part of the target protein. Taking into account this result, we recalculated the maximum capacity of the resins by summarizing the amount of protein obtained in the washing fraction at pH 5.5 and in the eluate fraction. The resulting capacity of Ruselect-P was 39.3, 31.9, and 39.0 mg of antibodies per 1 mL of sorbent for IgG1, IgG2, and IgG4, respectively, which, in comparison with MabSelect SuRe, is only ~10% lower for IgG1 and IgG4 but 23% lower for IgG2 (Figure 5). To avoid target protein losses in further experiments with the Ruselect-P resin, we used a washing buffer with a pH of 6.0 (the minimum value at which protein losses are insignificant).

Affinity chromatography plays a significant role in the removal of major impurities during the production of mAbs. So, the usage of special additional washes is a rational option at this stage to effectively eliminate impurities. We have determined the optimal conditions for the purification of antibodies of each subclass on the Ruselect-P and MabSelect SuRe resins, based on parameters such as the yield of IgG, chromatographic purity, and residual host cell protein (HCP) content in the eluate. The pH values of the elution buffer varied in the range of 3.4–4.4 in the experiments (Table 4). For the Ruselect-P resin, the effectiveness of washing with a 1 M NaCl solution at pH 7.3 and 10.0 was also studied (Table 5). A protein load of 25–30 mg of IgG per 1 mL of resin was used in all described experiments (approximately 80% of the maximum capacity for the corresponding antibody).

In order to find out the optimal pH value of the elution buffer, screening was initiated at pH 3.4, followed by stepwise pH increases until the yield began to decrease (for IgG2 on Ruselect-P, the starting pH value of 3.6 was used due to the low affinity of this antibody to the resin). The results showed that elution at pH 3.6 was optimal for all IgGs on the MabSelect SuRe resin, while on Ruselect-P, the optimal pH of elution increased in the following order: 3.6, 3.8, 4.0 (for IgG4, IgG1, and IgG2, respectively). This suggests a corresponding decline in the affinity of antibodies to the ligand (Table 4). It should be noted that, at the optimal pH value of the elution buffer, the product yield turned out to be higher compared to more acidic pH values, which is probably due to the fact that a lower pH provokes the formation of oligomeric IgG forms that interact more strongly with the resin and remain bound to the ligand. This may also explain the yields, which were found out to be above 100% relative to analytical determinations of HCCF productivity, in which MabSelect SuRe sorbent with elution at pH 3.2 was used.

It was also observed that there was a significant decrease in the amount of HCP impurities (up to 2.5 times) in the samples with a higher elution pH. The HCP concentration varies among different antibodies, depending on the specific cultivation conditions. The purity of IgG in the eluates from both resins was similar, but MabSelect SuRe showed slightly better efficiency in removing impurities.

The Ruselect-P resin also showed an excessive pH buffering effect: after a regeneration by 0.1 M NaOH solution, approximately 40 CV of phosphate-buffered saline was required to equilibrate the resin, which is eight times more than that used for equilibration of MabSelect SuRe in the same conditions. However, the introduction of an additional washing step, for example 1–2 CV of 100 mM acetic acid, could resolve this issue of the Ruselect-P buffering effect.

Having examined the influence of washing the Ruselect-P resin with a 1 M NaCl solution at different pH values on the purity of the product, we found that it strongly depends on the individual properties of the IgG and the nature of the associated impurities. For instance, the application of these washing buffers resulted in both a significant decrease in IgG_4_ yield and an increase in its chromatographic purity, with the HCP amount decreasing by almost 10-fold. As for IgG_2_, an effect was observed only on the HCP content (~3.5 times decrease). No impact on yield or purity was detected in the case of IgG_1_ (Table 5).

We also measured the capacity of the Ruselect-P and MabSelect SuRe resins at different RTs (2, 3, and 6 min; Figure 6A–C) for IgG1, IgG2, and IgG4 antibodies. As expected, an increase in the capacity in correlation with the residence time was observed in all experiments, but the increase was less than 10% in most cases. Only IgG4 exhibited a 14% increase in the Ruselect-P capacity at the 6 min residence time compared to the 2 min residence time.

In order to evaluate ligand leakage during chromatographic purification, the IgG2 antibody was isolated from HCCF on the RUselect-P resin. Having conducted 10 cycles of purification, the amount of protein A was measured in the eluate samples (Table 6).

The amount of protein A decreased by approximately four times, reaching 1.8 ng per 1 mg of IgG_2_ after 10 cycles of purification. Although this value is slightly higher than that for the MabSelect SuRe resin (~0.5–1 ng/mg), it should not be a serious problem in subsequent purification of the product.

## 4. Discussion

A new affinity resin with a ligand based on the mutant domain B of protein A was developed. The key feature of the proposed method is the use of the sortase A enzyme for ligand immobilization. We synthesized an affinity resin (RUselect-P) with a high DBC_10_ value (up to 50 mg of IgG_1_ per 1 mL of sorbent for bevacizumab), strong resistance to CIP processes, and reusability. The RUselect-P resin exhibited all the necessary characteristics (stability, capacity, purification efficiency) for its use in laboratory and industrial processes for the isolation and purification of antibodies (Table 7).

The features of the RUselect-P resin make it possible, in some cases, to use more gentle modes of antibody elution, facilitating further purification of the protein, which is especially important for the purification processes of therapeutic mAbs.

## 5. Conclusions

We have demonstrated the possibility of sortase A enzyme to immobilize recombinant protein A on the surface of matrices modified with primary amino groups. This greatly simplifies the synthesis of the resin. The existing methods of site-specific immobilization of protein A and its analogues are based on a chemical method of immobilization, for example through C-terminal cysteine. These methods include several stages of matrix activation and preparation, including the introduction of additional linkers. The approach described in this manuscript allows for the immobilization of the ligand in one stage and under mild conditions. We suggest that the presented method can be used not only for the protein A resin preparation but also for the development of various chromatography resins using LPXTG-containing protein ligands.

## 6. Patents

Miagkikh, I. V.; Stepanenko, V. N.; Tereshin, M. N.; Melikhova, T. D.; Eletskaya, B. Z. Recombinant analogue of B-domain of the BDPA-1 protein A; recombinant plasmid for the expression of the BDPA-1 protein; *Escherichia coli* producer strain producing the BDPA-1 protein; method for producing the BDPA-1 protein; and method for creating an affinity sorbent containing BDPA-1 or a fragment thereof as a ligand. Patent RU 2789032 C1. 2023.

## Data Availability

The data presented in this study are contained within the article.

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
