# Peer review of "Biocatalytic Method for Producing an Affinity Resin for the Isolation of Immunoglobulins"

_biomolecules, 2024, doi:10.3390/biom14070849_

Round 1

Reviewer 1 Report

Comments and Suggestions for Authors

This article presents the protocol for the synthesis and application of a novel affinity resin immobilized with a mutant form of protein A as a ligand with a high dynamic binding capacity (up to 50 mg IgG per 1 ml sorbent). In the presented protocol, the authors used the biocatalytic immobilization of the ligand on the matrix using the enzyme sortase A. In addition, the authors used a matrix with primary amino groups without modification, which greatly simplifies the synthesis process. The main advantage of the designed and synthesized resin is its high tolerance to 0.1 M NaOH treatment, maintaining its efficacy even after 100 cycles of binding, elution and purification.

The work was very well planned and carried out step by step. The research results are reliably described, the experimental part is complete and accurate.

The manuscript can be considered for publication upon completion of the following elements

1) The authors wrote: "The production method of recombinant BsrtA protein is described in patent RU Yeah. 2,789,032 C1, published on January 27, 2023". Was the obtained protein characterized by analytical methods before immobilization? It is worth writing something about this, of course without giving away details that cannot be given.

2) The authors emphasize the possibility of multiple uses of the resin. However, there were no elements of validation, such as inter-day (intra-day) detection efficiency, calculation of standard deviation, relative standard deviation, and student's test. This is an important element of this paper and should be considered.

3) Do the authors consider routine use of this method in laboratories? What are the future applications of this resin?

Author Response

Our team would like to express our sincere gratitude for your careful and thorough review of the article. Your comments have been very helpful in improving the quality of the content. We appreciate your valuable feedback.

Response 1: The authors wrote: "The production method of recombinant BsrtA protein is described in patent RU Yeah. 2,789,032 C1, published on January 27, 2023". Was the obtained protein characterized by analytical methods before immobilization? It is worth writing something about this, of course without giving away details that cannot be given.

Comments 1: Added to the text: The obtained protein was analyzed using analytical HPLC and SDS-PAGE methods. The purity was >90 % in both methods.

Response 2: The authors emphasize the possibility of multiple uses of the resin. However, there were no elements of validation, such as inter-day (intra-day) detection efficiency, calculation of standard deviation, relative standard deviation, and student's test. This is an important element of this paper and should be considered.

 Comments 2:

Added to the text: «Intraday and interday precisions for IgG1 bevacizumab were <1.7 % and <2.0% RSD (Relative Standard Deviation), respectively. The data were analyzed using Student’s t-test, P < 0.05 was considered significant.»

Response 3:

Do the authors consider routine use of this method in laboratories? What are the future applications of this resin?

Comments 3:

RUselect-A (cross-linked agarose matrix) and RUselect-P (polimer matrix)  resins are already used in several Russian pharmaceutical companies that specialize in mAbs production.

Reviewer 2 Report

Comments and Suggestions for Authors

Review comments:

This MS is a well-organized and well-written article describing the enzymatic conjugation of affinity ligand to a chromatography matrix. To improve its quality and reader friendliness, I recommend the following minor revisions:

1. SrtA recognition site, LPXTG, is written from C- to N-terminus direction. Would it be better to change its direction from N- to C? The left side of the Fig. 1 also shows the sequence from C- to N.

2. What is Yn in Fig.1? Provide the explanation in the text.

3. Several lines under section 2.5 (line 147-155) is duplicated with line #90-98.

4. [Line #238] “Six introduced single mutation…” Is it related with “Yn” in Fig. 1? It is better to specifically identify the mutations, i.e., amino acid residues for the mutations.

5. [Table 7] If available, include the data on pore size distribution or pore volume in the resin characteristics.

[END]

Author Response

Our team would like to express our sincere gratitude for your careful and thorough review of the article. Your comments have been very helpful in improving the quality of the content. We appreciate your valuable feedback.

Response 1: SrtA recognition site, LPXTG, is written from C- to N-terminus direction. Would it be better to change its direction from N- to C? The left side of the Fig. 1 also shows the sequence from C- to N.

Comments 1: Corrected.

Response 2: What is Yn in Fig.1? Provide the explanation in the text.

Comments 2: We introduced the explanation

Response 3: Several lines under section 2.5 (line 147-155) is duplicated with line #90-98.

Comments 3: Removed

Response 4: [Line #238] “Six introduced single mutation…” Is it related with “Yn” in Fig. 1? It is better to specifically identify the mutations, i.e., amino acid residues for the mutations.

Comments 4: Figure 1 illustrates the general mechanism of immobilization using the sortase A. Figure 2 shows the ligand that we have developed. The mutations used and detailed structure of the ligand can be found in patent RU 2 789 032 C1.

Response 5: [Table 7] If available, include the data on pore size distribution or pore volume in the resin characteristics.

Comments 5: We specified this parameter (70 microns) according to the manufacturer's specification from Nano Micro Technology. We do not have more detailed information.

Reviewer 3 Report

Comments and Suggestions for Authors

The manuscript by Tereshin et al. reports the development of an excellent affinity resin that captures IgGs by immobilizing a fusion protein (BsrtA), which has two domain Bs of protein A tandemly linked, on a support using sortase A. Therefore, the main focus of this paper is the development of BsrtA and its immobilization using sortase A. However, since BsrtA is patented, many details, including its development, are not described. In fact, the introduction explains that the focus of the paper is on immobilization using sortase A, as indicated by the inclusion of "biocatalytic method" in the title. Nevertheless, the use of BsrtA is an important point for the authors, and section 3.4 explains the strengths of BsrtA. Thus, the focus of this paper seems blurred. If the authors want to change the purpose of the paper, section 3.4 should report a comparison between direct immobilization of BsrtA instead of f MabSelect SuRe. Another major issue is the lack of a negative control in the immobilization with sortase A. To confirm that the immobilization was indeed performed by sortase A, a comparison with immobilization without sortase A should be provided.

Minor points:

L86-89: The paper describes what was done, but does not include the research objectives, working hypothesis, or motivations.

L100: Abbreviations should be explained prior to their first use.

L 125: The method of modifying the support with the linker should be described in the main manuscript, not in the Supplementary Information (SI-3.3).

SI-3.4: This information should also be provided in the main manuscript.

Section 2.4: The temperature used to immobilize BsrtA with sortase A is not mentioned.

L145-155: This section repeats what is already stated in the materials section and should be removed.

L178: What is CIP?

Section 3.1: Information about BsrtA should also be provided in the Introduction.

L244-246This part should be removed.

Table 3: The relationship between the loss in DBC10 and the amount of immobilization should be described.

L338: MabSelect SuRe needs to be explained.

Author Response

Our team would like to express our sincere gratitude for your careful and thorough review of the article. Your comments have been very helpful in improving the quality of the content. We appreciate your valuable feedback.

We conducted a direct comparison of the chemical and enzymatic immobilization methods for our patented protein (lines 254-257 and Table 2). We compared DBC10 values and CIP-resistance of the resins obtained using different methods. The chemical method of immobilization was not used further in the work, as it showed worse characteristics. In Section 3.4, we demonstrated the use of our developed resin in industrial process of mAbs production, which was also the purpose of this work. Therefore, we compared our product with commercial MabSelect resin. To ensure that the purpose of our work is not blurred, we’ve changed the last paragraph in the Introduction (L86-95).

As regards the negative control, we performed immobilization without sortase. The resulting resin exhibited zero DBC. Moreover, during the final stages of resin synthesis (198-201), according to the technology, the resin was washed with various solutions (PBS (pH 7.4); 100 mM sodium acetate (pH 4.5); PBS (pH 7.4); 100 mM NaOH; 50 mM Tris-HCl (pH 8.0); aqueous solution of 20% ethanol), which excludes non-specific ligand binding.

Response 1: L86-89: The paper describes what was done, but does not include the research objectives, working hypothesis, or motivations.

Comments 1: We’ve changed the last paragraph in the Introduction to the following:

Also added to introduction: «This research aimed to investigate approach for the immobilization of the protein A analog, with the objective of developing affinity resins with improved capacity and stability and evaluating their application in isolating monoclonal antibodies (mAbs). To achieve this, an engineered mutant protein A with a sortase A recognition site and a C-terminal cysteine was designed and patented. Two methods for site-specific immobilization were then compared: chemical attachment via the C-terminal cysteine residue and enzymatic conjugation using sortase A. The obtained sorbents were tested using purified antibody (bevacizumab) to determine the most effective approach to ligand immobilization. Then, the best sample was tested against two types of antibodies (bevacizumab and adalimumab) in comparison to the commercial resin MabSelect Sure. The properties of this resin were investigated under real conditions to purify the antibodies (types IgG1, IgG2, and IgG4) from the harvested cell culture fluid.»

Response 2: L100: Abbreviations should be explained prior to their first use.

Comments 2: BsrtA is the name of our ligand. We added a mention of this in the abstract.

Response 3: L 125: The method of modifying the support with the linker should be described in the main manuscript, not in the Supplementary Information (SI-3.3).

SI-3.4: This information should also be provided in the main manuscript.

Comments 3: We moved the SI 3 section from SI to the Materials and methods

Response 4: Section 2.4: The temperature used to immobilize BsrtA with sortase A is not mentioned.

Comments 4: The temperature was indicated in the section "2.7. Enzymatic immobilization of BsrtA using sortase A" (37 °C)

Response 5: L145-155: This section repeats what is already stated in the materials section and should be removed.

Comments 5: Removed.

Response 6: L178: What is CIP?

Comments 6: We added the explanation

Response 7: Section 3.1: Information about BsrtA should also be provided in the Introduction.

Comments 7: We provided some additional information.

«This research aimed to investigate approach for the immobilization of the protein A analog, with the objective of developing affinity resins with improved capacity and stabil-ity and evaluating their application in isolating monoclonal antibodies (mAbs). To achieve this, an engineered mutant protein A with a sortase A recognition site and a C-terminal cysteine was designed and patented. Two methods for site-specific immobili-zation were then compared: chemical attachment via the C-terminal cysteine residue and enzymatic conjugation using sortase A. The obtained sorbents were tested using purified antibody (bevacizumab) to determine the most effective approach to ligand immobiliza-tion. Then, the best sample was tested against two types of antibodies (bevacizumab and adalimumab) in comparison to the commercial resin MabSelect Sure. The properties of this resin were investigated under real conditions to purify the antibodies (types IgG1, IgG2, and IgG4) from the harvested cell culture fluid.»

Response 8: L244-246This part should be removed.

Comments 8: Removed

Response 9: Table 3: The relationship between the loss in DBC10 and the amount of immobilization should be described.

Comments 9: There is no direct relationship between the resistance to sanitization and the amount of immobilized ligand. The more ligand is immobilized, the more the resulting DBC, but the loss of DBC depends on the linker’s chemical structure. (L334-335)

Response 10: L338: MabSelect SuRe needs to be explained.

Comments 10: Added to the text: MabSelect SuRe (commercial protein A-based affinity resin).

Round 2

Reviewer 1 Report

Comments and Suggestions for Authors

The manuscript was improved. The manuscript may be accepted in its present form.

Author Response

Thank you for your edits on our manuscript.

Reviewer 3 Report

Comments and Suggestions for Authors

The revised manuscript is much improved. However, a major point has not been fully addressed: The authors has responded that they  performed negative control experiments without sortase A. However, the results are not shown in the manuscript. The authors should clearly mention that non-specific binding of BrstA does not occur by showing the data.

Author Response

Thank you for your edits on our manuscript.

We have added a paragraph about the negative control to the Materials and Methods section.

Round 3

Reviewer 3 Report

Comments and Suggestions for Authors

If "After washing, the sample was analyzed using the same method as the experimental sample, and the resulting DBC value was zero." means "Samples were washed until the DBC value became zero," I think the revised manuscript is acceptable.